# Bias A-head? Analyzing Bias in Transformer-Based Language Model Attention Heads

## Abstract

Transformer-based pretrained large language models (PLM) such as BERT and GPT have achieved remarkable success in NLP tasks. However, PLMs are prone to encoding stereotypical biases. Although a burgeoning literature has emerged on stereotypical bias mitigation in PLMs, such as work on debiasing gender and racial stereotyping, how such biases manifest and behave internally within PLMs remains largely unknown. Understanding the internal stereotyping mechanisms may allow better assessment of model fairness and guide the development of effective mitigation strategies. In this work, we focus on attention heads, a major component of the Transformer architecture, and propose a bias analysis framework to explore and identify a small set of **biased heads** that are found to contribute to a PLM's stereotypical bias. We conduct extensive experiments to validate the existence of these biased heads and to better understand how they behave. We investigate gender and racial bias in the English language in two types of Transformer-based PLMs: the encoder-based BERT model and the decoder-based autoregressive GPT model. Overall, the results shed light on understanding the bias behavior in pretrained language models.

## 1 Introduction

Transformer-based pretrained language models such as BERT (Devlin et al., 2018), GPT-2 (Radford et al., 2019), and large foundation models such GPT-3 (Brown et al., 2020), PaLM (Chowdhery et al., 2022), and LLaMA (Touvron et al., 2023) have achieved superior performance in many natural language processing (NLP) tasks (Adlakha et al., 2023; Gao et al., 2023; Li et al., 2023; Wei et al., 2023; Yao et al., 2023). However, since PLMs and foundation models are trained on large human-written corpora, they often encode undesired stereotypes towards different social groups, such as gender, race, or people with disabilities (Bender et al., 2021; Blodgett et al., 2020; Hutchinson et al., 2020). For example, GPT-2 has been shown to generate stereotypical text when prompted with context containing certain races such as African-American (Sheng et al., 2019). A stereotype is an over-simplified belief about a particular group of people, e.g., "women are emotional." Stereotyping can cause representational harms (Blodgett et al., 2020; Barocas et al., 2017) because it can lead to discrimination, prejudice, and unfair treatment of individuals based on their membership in a particular group (Fiske, 1998).

In order to design robust and accountable NLP systems, a rich and growing body of literature has investigated the stereotypes in PLMs from two perspectives. The first line of work aims to quantify the stereotypical biases. For example, May et al. (2019) propose a Sentence Encoder Association Test (SEAT), and Nadeem et al. (2021) develop the StereoSet dataset to assess if a PLM encodes stereotypes. The second line of work aims to propose de-biasing strategies that remove undesired stereotypical association biases from PLMs (Zhou et al., 2023; Guo et al., 2022; He et al., 2022; Kaneko & Bollegala, 2021). Similarly, foundations model also needs to be further aligned to alleviate its bias concern, using techniques such as Reinforcement Learning from Human Feedback (RLHF) (Ouyang et al., 2022). However, there are still gaps in understanding stereotypical biases in transformer-based language models. For bias assessment, while the common practice uses one score to quantify the model bias, it is unclear how the bias manifests internally in a language model. For bias mitigation, existing works are usually designed in an end-to-end fashion with a "bias neutralization" objective, but the inner-workings of the entire debiasing procedure remain a black-box. There is a need for in-depth analysis that uncovers how biases are encoded *inside* language models.

In this work, we propose a framework to analyze stereotypical bias in a principled manner.[1] Our main research question is, *how does bias manifest and behave internally in a language model?* Prior work in better understanding the internal mechanisms of deep neural networks has focused on specific model components. For example, we take inspiration from the seminal work of finding a single LSTM unit which performs sentiment analysis (Radford et al., 2017) and attributing types of transformer attention heads as "induction heads" that do in-context learning (Olsson et al., 2022). In this work, we focus on attention heads in pretrained language models. Attention heads are important because they enable transformer-based models to capture relationships between words, such as syntactic, semantic, and contextual relationships (Clark et al., 2019).

Our proposed framework begins by measuring the bias score of each Transformer self-attention head with respect to a type of stereotype. This is done by deriving a scalar for each attention head, obtained by applying a gradient-based head importance detection method on a bias evaluation metric, i.e., the Sentence Encoder Association Test (SEAT, May et al., 2019). Heads associated with higher bias scores are dubbed **biased heads**, and are the heads upon which we then conduct in-depth analyses.

In our analysis, we start by investigating how gender biases are encoded in the attention heads of BERT. We visualize the positions of biased heads and how they are distributed across different layers. To further verify that the identified biased heads indeed encode stereotypes, we conduct a counter-stereotype analysis by comparing the attention score changes between the biased heads and normal (non-biased) heads. Specifically, given a sentence containing a gender stereotype such as "women are emotional," we obtain its counter-stereotype "men are emotional." We then calculate the attention score change for the stereotypical word "emotion." Since the only difference between the original sentence and its counter-stereotype sentence is the gender-related word, we would expect significant score changes for those heads that encode biases, and minimal changes for those heads that do not encode biases. Our analysis on a large external corpus verifies that the attention score change of identified biased heads are statistically and significantly greater than that of the normal heads.

Later in the paper, we extend the analysis to investigate bias in the GPT model, as well as racial stereotype associated with Caucasians and African Americans. Moreover, we show that a simple debiasing strategy that specifically targets a small set of biased heads (by masking), which is different from previous end-to-end bias mitigation approaches that tune the entire PLM, yields a lower model bias performance with minimal disruption to language modeling performance.

In summary, this work makes two important contributions. First, we open the black-box of PLM biases, and identify biased heads using a gradient-based bias estimation method and visualizations, shedding light on the internal behaviors of bias in large PLMs. The proposed framework also contributes to the literature on understanding how PLMs work in general (Rogers et al., 2020). Second, we propose a novel counter-stereotype analysis to systematically study the stereotyping behavior of attention heads. As a resource to the research community and to spur future work, we will open-source the code used in this study.

## 2 BACKGROUND

### 2.1 MULTI-HEAD SELF-ATTENTION

Multi-head self-attention in Transformers is the fundamental building block for language models (Vaswani et al., 2017). In short, the self-attention mechanism allows a token to attend to all the tokens in the context, including itself. Formally, $head_{i,j}$ denotes the output of attention head $j$ in layer $i$., i.e., $head_{i,j} = Attention(Q_{i,j}, K_{i,j}, V_{i,j})$, where $Q_{i,j}$, $K_{i,j}$, and $V_{i,j}$ are learnable weight matrices. A language model usually contains multiple layers of Transformer block and each layer consists multiple self-attention heads. For example, BERT-base contains $L = 12$ layers of Transformers block, and each layer consists of $H = 12$ self-attention heads.[2]

---

[1]Throughout the paper, we use the term *bias* to refer to stereotypical bias.

[2]In this paper, we use <layer>−<head number> to denote a particular attention head, and both the layer index and head index start with 1. For example, the 12-th head in the 9-th layer in BERT-base model is denoted as 9-12.

The attention outputs are concatenated and then combined with a final weight matrix by extending the self-attention to multi-headed attention:

$$MultiHead_i(X_{i-1}) = \underset{j=1...H}{Concat} (head_{i,j}) \, W^O, \tag{1}$$

where $W^O$ serves as a "fusion" matrix to further project the concatenated version to the final output, and $X_{i-1}$ is the output from the previous layer.

## 2.2 STEREOTYPING AND REPRESENTATIONAL HARMS IN PLMS

A growing body of work exploring AI fairness in general, and bias in NLP systems in particular, has highlighted stereotyping embedded in state-of-the-art large language models – that is, such models represent some social groups disparately on demographic subsets, including gender, race, and age (Bender et al., 2021; Shah et al., 2020; Guo & Caliskan, 2021; Hutchinson et al., 2020; Kurita et al., 2019; May et al., 2019; Tan & Celis, 2019; Wolfe & Caliskan, 2021; Rozado, 2023). According to the survey of Blodgett et al. (2020), a majority of NLP papers on bias study representational harms, especially stereotyping. Our work is in line with the branch of research on exploring stereotypical bias in Transformer-based PLMs.

Prior work proposes several ways of assessing the stereotyping encoded in a PLM. A commonly used metric is the Sentence Encoder Association Test (SEAT) score, which is an extension of the Word Embedding Association Test (WEAT, Caliskan et al., 2017), which examines the associations in contextualized word embeddings between concepts captured in the Implicit Association Test (Greenwald et al., 1998). While the SEAT score provides a quantifiable score to evaluate the stereotyping in PLMs, it is unknown how such stereotypical associations manifest in PLMs.

To mitigate stereotyping and representational harms in PLMs, many different debiasing strategies have been proposed, including data augmentation (Garimella et al., 2021), post-hoc operations (Cheng et al., 2021; Liang et al., 2020), fine-tuning the model (Kaneko & Bollegala, 2021; Lauscher et al., 2021), prompting techniques (Guo et al., 2022), and Reinforcement Learning from Human Feedback (RLHF) (Ouyang et al., 2022). However, recent literature has noted several critical weaknesses of existing bias mitigation approaches, including the effectiveness of bias mitigation (Gonen & Goldberg, 2019; Meade et al., 2022), high training cost (Kaneko & Bollegala, 2021; Lauscher et al., 2021), poor generalizability (Garimella et al., 2021), and the inevitable degradation of language modeling capability (He et al., 2022; Meade et al., 2022). We believe that progress in addressing PLM bias has been inhibited by a lack of deeper understanding of how the bias manifests/behaves *internally* in the PLM. This paper aims to offer a perspective on this research gap.

## 3 ATTENTION HEAD BIAS ESTIMATION FRAMEWORK

Our proposed framework for attention head bias estimation measures the bias score of Transformer self-attention heads with respect to a focal/concerning bias (e.g., gender). We first introduce a new variable, the *head mask* variable, that exists independently in each attention head. We then discuss how this variable can be utilized to quantify the bias in each attention head.

### 3.1 HEAD MASK VARIABLE

Michel et al. (2019) propose a network pruning method that examines the importance of each self-attention head in a Transformer model. Given our interest in measuring the importance of each self-attention head with respect to a concerning bias, for each attention layer $i$ comprised of $H$ attention heads, we introduce a variable $m_i = [m_{i,1}, m_{i,2}, \ldots, m_{i,H}]'$ called the head mask variable that is multiplied element-wise with the output from each attention head in the $ith$ layer. This allows us to understand (and control) the contribution of each attention head to the model's final output:

$$MultiHead_i(X_{i-1}) = \underset{j=1,...,H}{Concat} (m_{i,j} \cdot head_{i,j}) \, W^O, \tag{2}$$

where $m_{i,j}$ is a scalar initialized with 1 in our implementations. In Equation 2, if $m_{i,j} = 0$, it signifies that the attention head $i$-$j$ is completely masked out from the language model, that is, it

contributes nothing to the model's final output. On the contrary, if $m_{i,j} = 1$, it is degenerated into its standard multi-head attention form as shown in Equation 1.

## 3.2 ESTIMATING BIAS FOR EACH ATTENTION HEAD

Next, we show how this head mask variable can be utilized to quantify biases for each attention head. Formally, let $X$ and $Y$ be two sets of target words of equal size, and let $A$ and $B$ be two sets of attribute words. Here, target words are those that should be bias-neutral but may reflect human-like stereotypes. For example, in the context of gender bias, target words include occupation-related words such as *doctor* and stereotyping-related words such as *emotional*, and attribute words represent feminine words (e.g., *she*, *her*, *woman*) and masculine words (e.g., *he*, *his*, *man*). We assume $X$ is stereotyped with $A$ (e.g., stereotype related to female) and $Y$ is stereotyped with $B$ (e.g., stereotype related to male) . Since we aim to measure how much stereotypical association is encoded in each of the attention heads, we directly use the absolute value of the Sentence Encoder Association Test score as the objective function, as follows:

$$\mathcal{L}_{|SEAT|}(X, Y, A, B) = \frac{|mean_{x \in X} s(x, A, B) - mean_{y \in Y} s(y, A, B)|}{std\_dev_{w \in X \cup Y} s(w, A, B)}, \tag{3}$$

where $s(w, A, B) = mean_{a \in A} cos(\overrightarrow{w}, \overrightarrow{a}) - mean_{b \in B} cos(\overrightarrow{w}, \overrightarrow{b})$ and $cos(\overrightarrow{a}, \overrightarrow{b})$ denotes the cosine of the angle between contextualized embeddings $\overrightarrow{a}$ and $\overrightarrow{b}$. [3] Therefore, the *bias score* of each attention head can be computed as:

$$b_{i,j} = \frac{\partial \mathcal{L}_{|SEAT|}}{\partial m_{i,j}}, \tag{4}$$

where a larger $b_{i,j}$ indicates head $i$-$j$ is encoded with higher stereotypical bias. Using the absolute value of the SEAT score as the objective function allows us to back-propagate the loss to each of the attention heads in different layers and quantify their "bias contribution." Therefore, if the bias score of an attention head is positive, it means that a decrease in the mask score from 1 to 0 (i.e., excluding this attention head) would decrease the magnitude of bias as measured by SEAT. In other words, the head is causing the SEAT score to deviate from zero and intensify the stereotyping (intensify either female-related stereotyping or male-related stereotyping or both). In contrast, an attention head with negative bias score indicates that removing the head *increases* the model's stereotypical association. Therefore, we define **biased heads** as those having positive bias scores, and the magnitude of bias score indicates the level of encoded stereotypes.

Our proposed attention head bias estimation procedure has several advantages. First, the procedure is model-agnostic. The objective function (i.e., $\mathcal{L}_{|SEAT|}$) can be easily customized/replaced to serve different purposes, providing flexibility for more general or specific bias analyses including different types of biases, datasets, and PLM model architectures. Second, it is only comprised of one forward pass (to compute $\mathcal{L}_{|SEAT|}$) and one backpropagation process (to compute $b_{i,j}$). Thus, it is computationally efficient for increasingly large foundation models. Third and critically, the bias score can quantify the importance of each attention head on the concerning bias. We later empirically evaluate the proposed bias estimation procedure, enhancing our understanding of stereotype in PLMs.

## 4 EXPERIMENTAL SETUP

**Gender and Racial Bias Word Lists:** Our analysis focuses on studying gender bias and racial bias, which are two of the most commonly examined stereotypes in PLMs. For gender bias, we employ attribute and target word lists used in prior literature (Zhao et al., 2018; Masahiro & Bollegala, 2019). In total, the gender attribute word list contains 444 unique words (222 pairs of feminine-masculine

---

[3]We use the outputs from the final layer of the model as embeddings. Each word in the attribute sets is a static embedding obtained by aggregating the contextualized embeddings in different contexts via averaging which has been shown as an effective strategy Kaneko & Bollegala (2021).

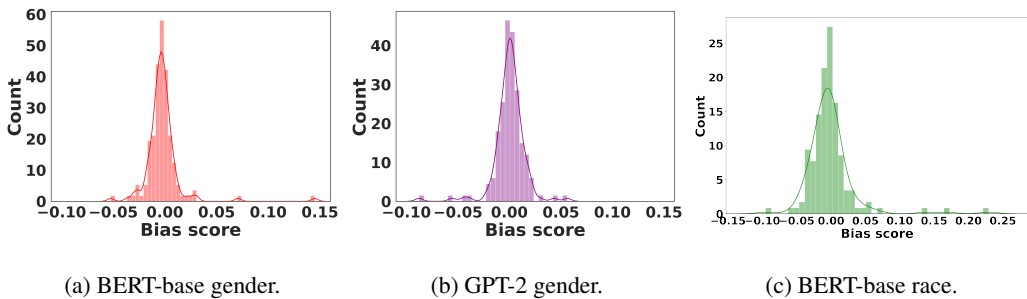

(a) BERT-base gender.  (b) GPT-2 gender.  (c) BERT-base race.

Figure 1: Bias score distributions for BERT-base gender (1a), GPT-2 gender (1b), and BERT-base race (1c).

words), and the target list contains 84 gender related stereotypical words.[4] For racial bias, we examine the stereotypical association between Caucasian/African American terms and stereotypical words. Specifically, we use the attribute word list and the target word list proposed in prior work (Manzini et al., 2019). The racial attribute word list contains 6 unique words (3 pairs of African-American vs. Caucasian words), and the target list contains 10 racial related stereotypical words.[5]

**External Corpus for Bias Estimation:** We use the News-commentary-v15 corpus to obtain contextualized word embeddings for PLMs and identify biased heads using the bias estimation method (Sec. 3.2). News-commentary-v15 corpust has often been used in prior PLM bias assessment and debiasing work (Masahiro & Bollegala, 2019; Liang et al., 2020).[6]

**PLMs:** We study the encoder-based BERT model and the decoder-based GPT model. For the BERT model, we consider BERT-base, which is comprised of 12 Transformer layers with 12 heads in each layer. For the GPT model, we consider GPT-2$_{Small}$ (Radford et al., 2019), which also consists of 12 Transformer layers with 12 attention heads in each layer. We implemented the framework and conducted experiments on an Nvidia RTX 3090 GPU using PyTorch 1.9. PLMs were implemented using the `transformers` library.[7]

## 5 ASSESSING GENDER BIAS IN BERT AND GPT

Prior literature has shown that PLMs like BERT and GPT exhibit human-like biases by expressing a strong preference for male pronouns in positive contexts related to careers, skills, and salaries (Kurita et al., 2019). This stereotypical association may further enforce and amplify sexist viewpoints when the model is fine-tuned and deployed in real-world applications such as hiring. In this section, we use the proposed method to assess gender bias in BERT and GPT-2.

### 5.1 DISTRIBUTION OF BIASED HEADS

There are 144 attention heads in BERT-base and GPT-2$_{Small}$; we obtain a bias score, $b_{i,j}$, for each of the attention heads. We visualize the bias score distribution in Figure 1a and Figure 1b respectively. It shows that most of the attention heads have a bias score that is centered around 0, indicating that they have no major effect on the SEAT score. Notably, there are several attention heads (on the right tail of the distribution curve) that have much higher bias scores compared to others. Moreover, GPT-2 contains more attention heads with pronounced negative bias scores than BERT, indicating that there are less biased attention heads in GPT-2.[8] In the ensuing analysis, we examine the biased heads, especially those with higher bias score values.

---

[4]https://github.com/kanekomasahiro/context-debias

[5]https://github.com/TManzini/DebiasMulticlassWordEmbedding/

[6]The dataset contains news commentaries, released for the WMT20 news translation task. We use the English data. https://www.statmt.org/wmt20/translation-task.html

[7]https://pypi.org/project/transformers/

[8]Relatedly, the SEAT score of GPT-2$_{Small}$ is 0.351 while that of BERT-base is 1.35.

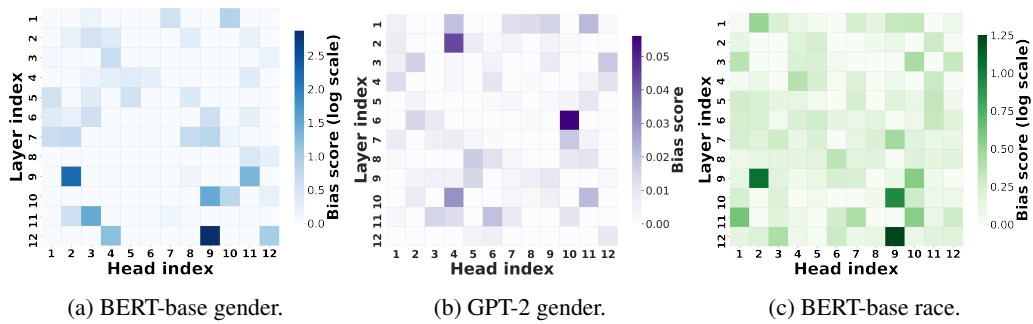

(a) BERT-base gender.    (b) GPT-2 gender.    (c) BERT-base race.

Figure 2: Attention head visualizations for BERT-base gender (2a), GPT-2 gender (2b), BERT-base race (2c). Note that negative bias scores are converted to zero for better visual illustration.

To understand the location of biased heads in BERT and GPT, we created a heatmap (Figure 2a and Figure 2b respectively) in which each cell represents a particular attention head, and the darker the color of the cell, the higher the bias score. Consistent with prior litarature (Kaneko & Bollegala, 2021), the identified biased heads appear across all layers.

## 5.2 COUNTER-STEREOTYPE EXPERIMENT

We now turn to evaluate if the identified biased heads - those attention heads with positive bias scores - indeed encode more stereotypical associations than non-biased attention heads with negative bias scores. We propose a *counter-stereotype experiment* for this purpose.

Although stereotyping in PLMs can be seen from the contextualized representations in the last layer, it is largely driven by how each token attends to its context in the attention head. By examining the attention maps (Clark et al., 2019) — the distribution of attention scores between an input word and its context words, including itself, across different attention layers — we can gain insight into how bias behavior manifests in PLMs.

We argue that we can gain insight into how bias behavior manifests in an attention head by examing how it assigns the attention score between two words. For example, given two sentences "women are emotional" and "men are emotional", since these two sentences have the exact same sentence structure except the gender attribute words are different, we should expect to see negligible attention score difference between the target word (emotional) and the gender attribute word (women, men). However, if an attention head encodes stereotypical gender bias that women are more prone to emotional reactions compared to men, there will be a higher attention score between "emotional" and "women" in the former sentence than that between "emotional" and "men" in the later sentence. In other words, simply substituting attribute words should not drastically change how the attention head works internally, unless the attention head is encoded with stereotypical associations. A running example is shown below.

**Running example:** We take an input text "*[CLS] the way I see it, women are more emtional beings...*" from the /r/TheRedPill corpus,[9] feed it into the BERT-base model, and visualize its attention maps, the distribution of attention scores (Clark et al., 2019), for the target word "*emotional*" at one biased head and one randomly sampled regular head in Figure 3.[10] Notably, for this biased head, the normalized attention score[11] between the target word *emotional* and the attribute word *women* is 0.0167. However, in the counter-stereotype example where *women* is substituted with *men*, the normalized attention score drops to 0.0073. All other things being equal, this head encodes more stereotypical associations. On the other hand, for the unbiased head, the change between attention score is negligible.

---

[9]/r/TheRedPill dataset contains 1,000,000 stereotypical text collected from the Reddit community (Ferrer et al., 2021).

[10]Note that for clarity, we do not display the attention with regards to special tokens (e.g., `[CLS]`, `[SEP]`) and punctuation (e.g., comma, period).

[11]The raw attention score is normalized using the min-max method, and the attentions to special tokens (i.e., [CLS] and [SEP]) and punctuation are excluded.

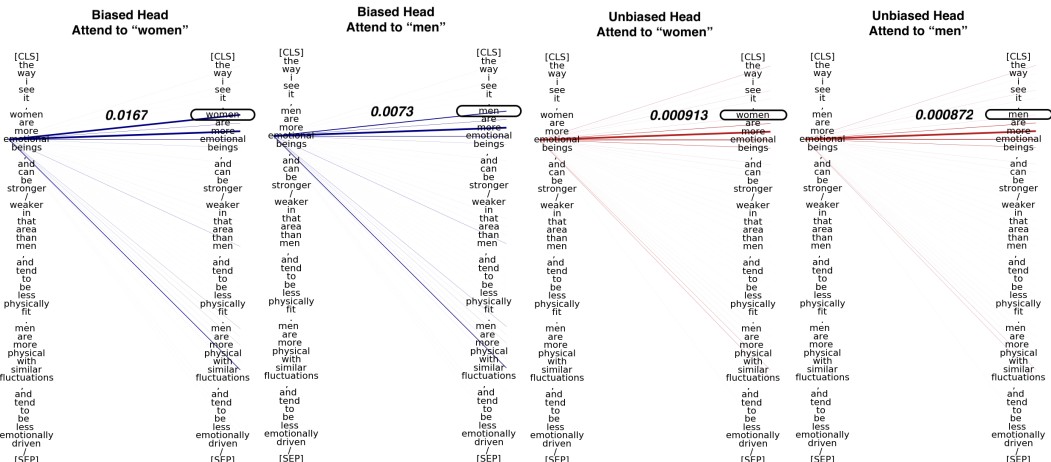

Figure 3: A running example for the counter-stereotype experiment. The four plots show the attention score (the boldface number) in the original sentence and the counter-stereotype sentence of a biased head (left two figures) and an unbiased head (right two figures). In this example, the target word is "emotional". The edge thickness is associated with its normalized attention score. BERT-base model is used in this example.

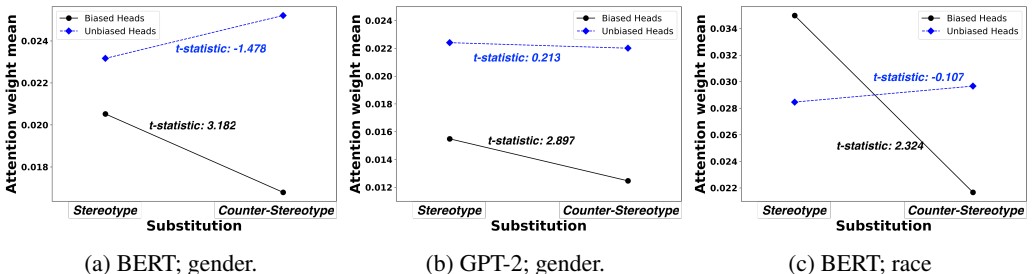

(a) BERT; gender.          (b) GPT-2; gender.          (c) BERT; race

Figure 4: Quantitative counter-stereotype experiments.

It is worth noting that the absolute value of the attention score does not necessary indicate the significance of bias. This is because the some attention heads may indeed be "gender" heads that associate high weights between gender words and target word, which could be very useful for context such as correference resolution. Therefore, to account for this, we measure the *difference* of attention score between a stereotype association (e.g., *women* and *emotional*) and a counter-stereotype association (e.g., *men* and *emotional*).

**Quantitative counter-stereotype analysis:** To assess the bias in biased heads more systematically and quantitatively, we conduct the counter-stereotype analysis using a large sample of sentences. The detailed steps are as follows.

**Step 1: Form a stereotype dataset.** We first obtain a set of sentences from TheRedPill corpus, where each sentence contains exactly one attribute word (e.g., "women") from our predefined word lists and one of its associated stereotypical target word (e.g., "emotional"). Note that this set of sentences could contain both women-related and men-related stereotype. We denote this dataset as $\mathcal{S}_{orig}$.

**Step 2: Form a counter-stereotype dataset.** We then construct a *counter-stereotype* dataset by replacing the attribute word (e.g., "women") with its counterpart (e.g., "men"), with all other words in the sentence unchanged, for each example in $\mathcal{S}_{orig}$. For example, given an original sentence "women are emotional," the counter-stereotype sentence would be "men are emotional." We denote this dataset as $\mathcal{S}_{counter}$. Note that sentences in $\mathcal{S}_{orig}$ and $\mathcal{S}_{counter}$ are paired, and the only difference in the paired sentences is that the stereotype related attribute words are different.

**Step 3: Examine attention score difference and statistical significance.** For Head $i$-$j$ (the $j$-th head in the $i$-th layer), we calculate the attention score that the target word has on the attribute word for each of the sentences in $s \in \mathcal{S}_{orig}$, which we denote as $w^s_{[i-j]}$. Similarly, we calculate the attention score for each of the counter-stereotype sentences $s\prime \in \mathcal{S}_{counter}$, which we denote as $w^{s\prime}_{[i-j]}$. We measure the attention score change after the attribute word substitution as $d^s_{[i-j]} = w^s_{[i-j]} - w^{s\prime}_{[i-j]}$. We then conduct a one-tail t-test to examine the null hypothesis that $d^s_{[i-j]}$ equals to zero. If the examined focal attention head encodes stereotypical bias, we would see that $d^s_{[i-j]}$ is significantly greater than zero and thus reject the null hypothesis.

The counter-stereotype experiment results are presented in Figure 4a (BERT) and Figure 4b (GPT) respectively. For BERT, we can see that for the biased heads, whose bias score is positive, the average attention score in $\mathcal{S}_{orig}$ is statistically higher than that in $\mathcal{S}_{counter}$ ($t$-stat $= 3.182$, $p$-value $< 0.001$, $N = 500$). However, the average attention score difference in the regular heads are not statistically significant ($t$-stat $= -1.478$, $p$-value $= 0.93$, $N = 500$), indicating that there is no significant change of attention score. The results are similar for GPT. The average attention score of biased heads in GPT is statistically higher in the original group than in the counter-stereotype group ($t$-stat $= 2.897$, $p$-value $< 0.005$, $N = 500$). However, there is no statistical significance between the original group and the counter-stereotype group for the regular heads ($t$-stat $= 0.213$, $p$-value $= 0.42$, $N = 500$). Taken together, the counter-stereotype experiment validates that the attention heads we identify as biased heads indeed encode stereotypical biases.

It should be noted that our counter-stereotype experiment differs from StereoSet (Nadeem et al., 2021), which incorporates human-annotated stereotype and counter-stereotype sentences. In StereoSet, the examples of stereotype and counter-stereotype are represented by completely different sentences. In contrast, our counter-stereotype examples are constructed by altering only the attribute words (such as those related to gender), while the overall sentence context remains unchanged. This method enables us to examine how the attention score of a specific attention head changes in a controlled manner.

# 6 ADDITIONAL ANALYSIS

## 6.1 ASSESSING RACIAL STEREOTYPING

In this section, to demonstrate our bias analysis framework is also applicable to other types of biases beyond gender bias, we apply our framework to examine racial bias between Caucasian/African American terms and racial related stereotypical words such as criminal, runner, etc. In the following experiment, we use BERT-base as the underlying PLM.[12]

We visualize the bias score distribution and heat map in Figure 1c and Figure 2c respectively. Much like the distribution of gender bias in BERT, we observe several heads with significantly higher bias scores. Moreover, the biased heads appear across all layers; some of the highest scores are distributed in the higher layers.

We conduct a counter-stereotype experiment to validate the identified racial biased heads. Similar to the counter-stereotype experiment step for gender bias analysis, we first obtain a set of sentences from the Reddit corpus that contains both the racial attribute words (such as "black") and stereotypical words (such as "criminal"). Then we measure the attention score change in a sentence and its counterfactual by replacing an attribute word to its counterpart word (such as "white"). Figure 4c shows that for the bias heads, the average attention score is significantly lower in the counter-stereotype group than in the original group, indicating these heads encode stronger racial stereotype associations ($t$-stat $= 2.324$, $p$-value $< 0.05$, $N = 500$). In contrast, for the unbiased heads group, there is no statistical difference in the original sentences and their counter-stereotypes ($t$-stat $= -0.107$, $p$-value $= 0.54$, $N = 500$).

## 6.2 UNDERSTANDING DEBIASING THROUGH THE LENS OF BIASED HEADS

Existing bias mitigation approaches are usually designed in an end-to-end fashion and fine tune *all model parameters* with a bias neutralization objective or a bias neutral corpus. For example,

---

[12]The results are similar for GPT model, and are omitted for space considerations.

Attanasio et al. (2022) propose to equalize the attention probabilities of all attention heads, and counterfactual data augmentation debiasing (CDA) proposes to pretrain a language model with a gender-neutral dataset (Zmigrod et al., 2019). In this sub-section, we use the scores from our bias analysis framework to shed light on possible application of biased heads for bias-mitigation.

We examine a different debiasing strategy that specifically targets on a set of attention heads. As an initial exploration of targeted debiasing, we examine a simple strategy, called ***Targeted-Debias***, that masks out top-K attention heads that have the largest bias score (**Top-3**). In addition, we also examine an opposite targeted debiasing that masks out K attention heads with the most negative bias score (**Bottom-3**). Moreover, we mask out all attention heads with a positive bias score (**All**) (in the case of gender bias in BERT, there are 45 attention heads with a positive bias score).

To benchmark the performance of Targeted-Debias, we consider ***Random-Debias*** that randomly masks out K out of BERT-base's 144 heads. To evaluate the impact of masking out attention heads, we assess the model's bias using SEAT score, and we also evaluate the model's language modeling capability using *pseudo-perplexities* (PPPLs)[13] (Salazar et al., 2020), and model's Natural Language Understanding (NLU) capability on the GLUE tasks (Wang et al., 2018).

The main debiasing results are presented in Table 1a. We can see that Targeted-Debias (Top-3) achieves the best performance among the three debiasing strategies: it has the lowest SEAT and lowest PPPL scores. Compared to the two versions of Targeted-Debias (Top-3 vs. All(45) ), masking out more biased heads does not further lower SEAT, but does significantly worsen the language modeling performance (4.16 vs. 5.75). The Top-3 Targeted-Debias only slightly increases BERT's PPPL from 4.09 to 4.16. Interestingly, we can see that targeting on the anti-biased heads (Bottom-3) increases the overall model bias. Random-Debias, which randomly masks out attention heads, actually exacerbates model bias. We posit that this result makes sense, given that if random heads are removed, those biased heads that remain will have their bias amplified. The GLUE task results appearing in Table 1b show similar trends as the language modeling task. That is, masking out the top-3 biased heads achieves comparable NLU performance to the original BERT-base model, while masking out all biased heads significantly worsens model performance. Taken together, it is encouraging that a simple debiasing strategy, targeting a small set of highly biased heads, can reduce PLM bias without affecting language modeling and NLU capability.

| Targeted Debiasing strategy | | Evaluation metric | |
|---|---|---|---|
| | | SEAT | PPPLs |
| BERT-base | | 1.35 | **4.09** |
| Targeted-Debias | Top-3 | **1.21** | 4.16 |
| | Bottom-3 | 1.39 | 4.20 |
| | All | 1.21 | 5.75 |
| Random-Debias | 3 | 1.36 | 4.13 |
| | All | 1.46 | 5.80 |

(a) Targeted debiasing.

| Task | Metric | Result | | |
|---|---|---|---|---|
| | | 0 (Full) | Top-3 | All |
| RTE | Accuracy | **0.6905** | 0.6748 | 0.6452 |
| SST-2 | Accuracy | 0.9297 | 0.9308 | 0.9185 |
| WNLI | Accuracy | 0.5506 | **0.5818** | 0.5298 |
| QNLI | Accuracy | **0.9154** | **0.9154** | 0.9066 |
| CoLA | Matthews corr. | 0.5625 | **0.5702** | 0.5584 |
| MRPC | F1 / Accuracy | 0.8701 / 0.8266 | **0.8748** / **0.8277** | 0.8729 / 0.8220 |
| QQP | F1 / Accuracy | **0.8829 / 0.9129** | 0.8823 / 0.9128 | 0.8796 / 0.9105 |
| STS-B | Pearson / Spearman corr. | 0.8862 / 0.8847 | **0.8875 / 0.8847** | 0.8817 / 0.8782 |
| MNLI | Matched acc. / Mismatched acc. | 0.8394 / 0.8406 | **0.8454 / 0.8518** | 0.8380 / 0.8422 |

(b) GLUE benchmark.

# 7  CONCLUSION AND DISCUSSION

In this work, we present an approach to understand how stereotyping biases are encoded in the attention heads of pretrained language models. We infer that the biases are mostly encoded in a small set of biased heads. We further analyze the behavior of these biased heads, by comparing them with other regular heads, and confirm our findings. We also present experiments to quantify gender bias and racial bias in BERT and GPT. This work is among the first work aiming to understand how bias manifests internally in PLMs. Previous work has often used downstream tasks or prompting to examine a PLM's fairness in a black-box manner. We try to open up the black-box and analyze different patterns of bias. In doing so, we strengthen our understanding of PLM bias mechanisms. Future work can apply our method to assess concerning biases in increasingly large foundation models such as GPT-3 and LLaMA. Overall, our work sheds light on how bias manifests internally in language models, and constitutes an important step towards designing more transparent, accountable, and fair NLP systems.

---

[13]Performed on the test split of "wikitext-2-raw-v1" accessible through `https://huggingface.co/datasets/wikitext`.

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
