# OpenReview forum: "Bias A-head? Analyzing Bias in Transformer-Based Language Model Attention Heads"
_ICLR.cc/2024/Conference — ICLR 2024 Conference Withdrawn Submission_

### Official Review · Reviewer_ELVt · 2023-10-24

**Soundness:** 2 fair
**Presentation:** 3 good
**Contribution:** 2 fair
**Rating:** 3
**Confidence:** 3

**Summary:**

The paper proposes a bias analysis framework to explore and identify a small set of biased heads that are found to contribute
to a PLM’s stereotypical bias. Extensive experiments are conducted to validate the existence of these biased heads, and on its basis, investigate gender and racial bias in the English language in BERT and GPT.  The results shed light on understanding the
bias behavior in pretrained language models.

**Strengths:**

- The paper targets at identifying bias in pre-trained language models, a hot NLP topic.

- The paper is written with a high degree of clarity. The research is well motivated; the proposed framework is presented in an easy-to-understand manner; the experiments are organized in a logical order.

- The experiments are comprehensive in that it only includes validates the sensibility of the proposed bias estimation framework, but also touches upon the analysis of the other types of bias and debiasing approaches with the proposed framework.

**Weaknesses:**

- The overview on probing stereotypical biases in BERT-like models is slim. Many recent works along this research direction are missing, e.g., Stereo type and Skew: Quantifying Gender Bias in Pre-trained and Fine-tuned Language Models, EACL 2021.

- Even though the proposed framework can be used to evaluate any transformer-based models, the evaluation only involves BERT-base and GPT-2-small, undermining the value of the findings.

- In the counter-stereotype experiment, the statistical test is performed on the average attention score of all biased and unbiased heads which, in my opinion, largely weakens the validity of the argument that the deemed biased heads encode stereotypical biases. Instead, the t-test should be performed on a per-head manner, and a box plot of the t-values of the biased heads and unbiased heads should be plotted to show the distribution of t-values in both classes.

**Questions:**

- What does not the last row in table (a) (Random-Debias, All) standard for?

- Why are top-K and bottom-K heads masked out with K=3 in the last experiment? What are the results for other values of K? In order to achieve reduced PLM bias without affecting the model capability, should the number of masked out heads chosen based on the obtained bias scores in practice?

---

### Official Review · Reviewer_fmfX · 2023-11-01

**Soundness:** 2 fair
**Presentation:** 3 good
**Contribution:** 3 good
**Rating:** 6
**Confidence:** 4

**Summary:**

This paper studies the relationship between attention heads in
transformer-based models and stereotypical bias (for gender and race
as use-cases). The paper shows that attention heads contribute
differently to bias scores that measure intrinsic bias based on the
SEAT test and stipulates that attention heads that contribute more are
probably responsible for bias in LMs. To further analyze the
difference in attention heads wrt bias, the paper analyzes
counterfactual pairs of sentences that differ only in the gender group
and shows that the heads attend more between gendered words and their
corresponding sterotypical words (e.g., higher attention between
"women" and "emotional" than "men" and "emotional"). Last, masking the
top-3 most biased heads leads to slight decrease in bias scores when
compared to removing the least biased heads or 3 heads at random.

**Strengths:**

Analysis of the relationship between attention heads and stereotypical bias for gender and race

**Weaknesses:**

Bias is quantified using intrinsic measures of bias that have been shown not be correlated with bias in downstream tasks

While I find the analysis of attention heads wrt bias interesting, I
am not sure whether analyzing intrinsic measures of bias is useful or
impactful. There have been several papers that show issues with
intrinsic bias measures: they are not robust and there is little to no correlation with bias measured for a downstream task. I recommend some of the following papers:

https://aclanthology.org/2022.trustnlp-1.7.pdf
shows how simple rephrasing of sentences with different lexical choices but the same semantic meaning lead to widely different intrinsic bias scores

https://aclanthology.org/2021.acl-long.150.pdf
shows that intrinsic bias measures do not correlate with bias measured at the NLP task level

https://aclanthology.org/2022.naacl-main.122/
describes more issues related to bias metrics

https://aclanthology.org/2021.acl-long.81/
lists several issues with current datasets/benchmarks for bias auditing

**Questions:**

In the light that there is no or little correlation between intrinsic bias measures and bias observed in a downstream task, how do you think the analysis of bias in attention heads is useful for downstream tasks?

In my opinion, a similar analysis of the attention heads could be performed in the context of a downstream task and it would be stronger and more relevant/impactful. This paper (unpublished) is addressing a similar analysis in the context of downstream tasks: https://arxiv.org/abs/2305.13088

---

### Official Review · Reviewer_6KTe · 2023-11-03

**Soundness:** 3 good
**Presentation:** 3 good
**Contribution:** 2 fair
**Rating:** 5
**Confidence:** 4

**Summary:**

This paper proposes a framework to analyze stereotypical bias by identifing biased attention heads, and sheds light on the behavior of transformer-based language models. The recognition of biased attention head is realized by deriving a scalar for each attention head, and then applying a gradient-based head importance detection method on a bias evaluation metric. The experiment findings suggest that attention heads play a crucial role in encoding stereotypical biases in pretrained language models, and identifying and mitigating these biases can improve the fairness and inclusivity of natural language processing applications.

**Strengths:**

1. The proposed framework provides a principled and systematic approach to analyzing stereotypical bias in transformer-based language models.
2. The proposed method provides a flexible and model-agnostic way to estimating the bias contribution of each attention head.
3. The proposed method sheds light on the internal mechanisms of stereotypical biases in pretrained language models, which can inspire future research on improving model fairness and accuracy.

**Weaknesses:**

1. The proposed method only focuses on gender and racial bias, and may not be applicable to other types of biases (especially when we can hardly obtain distinct A/B or X/Y groups).
2. The authors suggest that we can mitigate the biases by removing or modifying the biased attention heads while preserving the model's performance on downstream tasks, but only GLEU results are reported. How masking biased attention heads affects performance on generative tasks also requires further exploration.
3. Although the analysis method and its finding on the bias analysis are valuable to some extent, this paper lacks of their applications and thus their real contributions to the community are still unclear.

**Questions:**

N/A